

# Ground beetles (Coleoptera; Carabidae) as an indicator of ongoing changes in forest habitats due to increased water retention

Emilia Ludwiczak, Mariusz Nietupski and Agnieszka Kosewska

Department of Entomology, Phytopathology and Molecular Diagnostics, University of Warmia and Mazury in Olsztyn, Olsztyn, Poland

## ABSTRACT

This study, concerning the epigeic fauna of carabid beetles (Coleoptera; Carabidae), was conducted in the north-east of Poland, in an area which is part of the Dąbrówka Forest Subdistrict and has been included in the "Small water retention program for the Province of Warmia and Mazury in 2006–2015". The purpose of the study was to assess the impact of the water retention implemented within the framework of the above program on assemblages of ground beetles. These insects are highly sensitive to any anthropogenically induced transformations. This analysis was based on the interactions among the analyzed insects caused by changes occurring in their habitat. During the 3-year study, 5,807 specimens representing 84 species were captured. The water storage had a significant influence on the structure of the Carabidae assemblages. Before the earthworks were constructed for the project, the beetle assemblages had comprised a large group of xerophilous species, whereas after the small retention reservoirs had been created, an increase in the contribution of hygrophilous species was noticed. The results indicate that the retention works cause alterations in the water and environmental conditions of the habitats, and thereby effect changes in the composition of Carabidae assemblages. Moreover, modification in water relations within a habitat causes long-term changes in the structural and functional diversity of the beetles.

## INTRODUCTION

Water, as the principal component modeling the natural environment, plays a significant role in shaping forest ecosystems, in the sense of being both a habitat-forming element and a factor which ensures the stability, sustainability and diversity of habitats (*Pierzgalski, 2008*; *Blumenfeld et al., 2010*; *Koralay & Kara, 2018*). Any aberrations from the natural water regime, such as periodic water deficits or excesses, are events that have an adverse effect on the whole ecological system (*Rulik & White, 2019*). Such irregularities can be a consequence of erroneously implemented water retention or naturally occurring hydrological and meteorological processes (*Liberacki & Szafrański, 2013*; *Miler et al., 2013*; *Kędziora et al., 2014*).

As mentioned by *Mioduszewski (2010)*, in compliance with the Framework Water Directive of the European Union and because of the occurrence of water deficits in the

Corresponding author
Emilia Ludwiczak,
emilia.ludwiczak@uwm.edu.pl

entire European Union, all member countries are obligated to maintain an inventory and to protect ecosystems that have an impact on the shaping of proper water balance in nature. They are also required to implement measures to counteract water deficits by improving the retention capacity of the biosphere. In order to take full advantage of forest ecosystems, complex actions are undertaken to decelerate the circulation of water in the catchment while preserving the natural landscape (*European Environment Agency, 2015*). Such measures, called forest retention, are modeled after natural processes occurring in the natural environment (*Gustafsson et al., 2012*). By restoring the natural retention properties of ecosystems, and thereby improving the availability of water, it is possible to enhance the diversity of flora and fauna in the habitat (*Hansson et al., 2005*; *Nietupski, Kosewska & Ciepielewska, 2007*; *Janusz et al., 2011*; *Kosewska & Nietupski, 2015*).

The modifications which occur in areas where water retention has been introduced can be assessed by analyzing the responses of living organisms. Both intra- and inter-species structures of dependencies between organisms and the area they inhabit provide valuable information about the natural environment (*Mc Geoch, 1998*; *Gerlach, Samways & Pryke, 2013*). Effective indicator species used in the monitoring of the natural environment are beetles (Coleoptera; Carabidae) living on the surface of the earth (epigeic) (*Tőzsér et al., 2019*). These insects, as bioindicators of the condition of the environment, are characterized by high sensitivity to changing habitat conditions, especially changes in moisture content. This means that their observation can provide specific data about the current state of the ecosystem in which they live (*Rainio & Niemelä, 2003*; *Avgin & Luff, 2010*; *Koivula, 2011*; *Kotze et al., 2011*, *Bednarska et al., 2018*).

The aim of this study was to answer the following research hypotheses: (1) long-term modifications of the habitat water relations affect changes in ground beetle diversity, causing the disappearance of xerophilous species and the increase of hygrophilic species, (2) the designated transects differ in quantitative and qualitative values for the carabidofauna caught, (3) retention works cause the disappearance of rare Carabidae species found on the Red List of endangered animals in Poland (*Pawłowski, Kubisz & Mazur, 2002*).

## MATERIALS AND METHODS

### Study area

The study was completed in north-eastern Poland, in the Province of Warmia and Mazury, in the Dąbrówka Forest Subdistrict (UTM DE 66). Letter from the Olsztyn Forest District confirming the consent to make the area available, sent to the PeerJ publisher.

Entomological observations were carried out on an area formerly used for farming and then afforested and turfed. Two study sites were selected, characterized by different moisture conditions and located at a different distance to retention water reservoirs ('a' and 'b') created as part of the Program of small retention for the Province of Warmia and Mazury in 2006–2015 (Fig. 1) (www.sporol.warmia.mazury.pl). The first study site was located 128 m from a forest road, 36 m from retention water reservoir 'a', and 62 m from retention water reservoir 'b'. The second study site was situated at a distance of 100 m

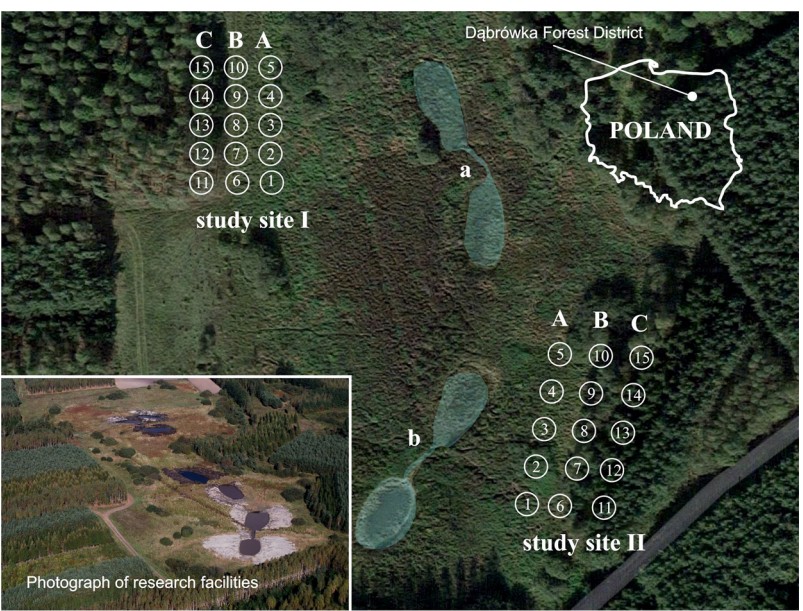

**Figure 1 Approximate localization of the study sites (I, II), water retention reservoirs (a, b) and the transects set up in the research (A–C).** Source: www.google.pl/maps, Olsztyn Forest District.

from the first, 47 m from the forest road, 39 m from reservoir 'a' and 15 meters from reservoir 'b' (*Forest Data Bank, 2019*).

Both study areas were characterized by similar habitat conditions. Three transects were chosen at each study site: transect A–on a waterlogged peat meadow, with the highest moisture content; transect B–an ecotone between the peat meadow and a forest, located 20 m away from transect A, and characterized by lower water abundance; transect C–situated in a pine forest, 20 m away from transect B, with the lowest moisture content.

The area covered by extensive meadow was dominated by peat soils of transitional peatland with the soil moisture content classified as a very wet, and very acid reaction. At both study sites, the meadow was adjacent to a 10-year-old mixed coniferous forest with the dominant species being common pine (*Pinus sylvestris* L.). This area was underlain mostly by Brunic Arenosols with the soil moisture content classified into the category: fresh wet (*Forest Data Bank, 2019*).

## Data collection

The water retention project in the Dąbrówka Forest Subdistrict within the framework of the program mentioned above started in 2010 with a design phase, and was continued in the form of earthworks in 2011. The comprehensive project consisted of new water retention reservoirs (www.olsztyn.olsztyn.lasy.gov.pl).

Studies on beetles of the Carabidae family were conducted in 2009, 2012 and 2013. The first entomological observations were made prior to the retention works connected with the project's implementation. The main purpose of the investigations carried out in 2009 was to make an assessment of the original state of Carabidae assemblages in the area designated for the project. The research continued in 2012 and 2013 after the

hydro-technical works had been completed, aimed at providing information on the extent to which the project affected the presence of the analyzed beetles.

In 2009, the field research began on 21 April and finished on 22 September (153 days). In 2012, the field research was carried out from 27 April to 26 October (181 days), and in 2013 it lasted from 25 April to 9 October (166 days).

In 2009, traps were placed at study site I, linearly, in three transects (A–C). In each transect, there were five traps spaced at 10 m (15 traps in total) (Fig. 1). The position of research study site I in 2009 was chosen intuitively as the exact location of the planned retention reservoirs was not known at that time. After the earthworks had been completed, the entomological observations were continued (in 2012 and 2013) in the area chosen in 2009. However, when the retention changes had been made, the research area was expanded by adding a second research study site, located to the east of the retention reservoirs. In 2012 and 2013, traps were dug at both study sites, 15 traps in each, set up in three rows and at the same 10-m distance as in 2009 (30 traps in total) (Fig. 1).

A modified Barber traps method was used for beetle capture, caught into containers with a capacity of 500 ml and a height of 12 cm (Barber, 1931). The containers were filled up to 1/3 with a solution of ethylene glycol including a small amount of detergent to decrease the surface tension. The traps were dug in level with the ground surface and emptied regularly every 14 days, collecting the entomological material and replacing the preserving solution.

Evaluation of meteorological conditions was based on measurements and observations carried out at the Meteorological Station in Olsztyn. Values of the mean annual air temperature during the 3 years of the study fluctuated only a little (7.7 °C in 2009, 7.5 °C in 2012 and 7.8 °C in 2013). The average sum of atmospheric precipitation during the 3 years of the study was 629 mm (604 mm in 2009, 700 mm in 2012, 582 mm in 2013) (Central Statistical Office in Poland, 2010, 2013, 2014).

## Data analysis

Species diversity and abundance of captured beetles were expressed in real numbers of caught specimens and species. The most popular tool, i.e., the Shannon index (Shannon & Weaver, 1949), was used to evaluate the biodiversity of the beetles, while the assessment of any deviations was based on mean individual biomass (MIB) (Szyszko et al., 2000).

The ecological analysis was carried out using the following indicators: trophic group, phenology, hygropreferences and habitat types (trophic group-(hemizoophages (Hz), small zoophages (Sz)): with a body length of no more than 15 mm, large zoophages (Lz): with a body length over 15 mm) (Leśniak, 1985; Aleksandrowicz, 2004; Kosewska, Nietupski & Ciepielewska, 2007), phenology-(spring breeders (Sb), autumn breeders (Ab)) (Larsson, 1939; Thiele, 1977), hygropreferences-(xerophilous (Xe), mesophilous (M), hygrophilous (H)) (Thiele, 1977; Aleksandrowicz, Pakuła & Mazur, 2008), habitat types-(forest (F), open area (Oa), peat bog (Pb), eurytopic (Eu)) (Aleksandrowicz, 2004; Kosewska, Nietupski & Ciepielewska, 2007).

To determine the significance of differences between the basic parameters describing the biodiversity of the Carabidae assemblages in the analyzed transects and years, the

generalized linear model (GLM) was used, which helped to determine the *p* values using Statistica 13.1 software (StatSoft, Inc., Tulsa, OK, USA). A test of the significance of effects comprised in the model was carried out according to Wald's statistics.

Jackknife two estimator was used for abundance data (using EstimateS v. 9.1.0 statistical software) and the species accumulation curves, were calculated to access the adequacy of the sampling efficiency (*Zahl, 1977*; *Colwell, 2005*).

Assessment of the similarity of the Carabidae assemblages in the examined transects and the years of the study was made with the help of a non-metric multidimensional scaling (NMDS), using Morisita's measure of similarity. Assessment of the significance of differences between the analyzed assemblages in the NMDS method was carried out using the ANOSIM non-parametric statistical test (*Kenkel & Orloci, 1986*; *Clarke, 1993*). Impact of environmental variables (year, transect) on the ecological groups of Carabidae was determined using redundancy analysis RDA (*Ter Braak & Šmilauer, 1998*). This method was chosen based on an analysis of the data distribution, which was linear (SD = 1.58).

Each trap was used as a replication for GLM analysis (mean values of abundance, species richness, mean individual biomass MIB) and for species accumulation curves. For NMDS and RDA analysis the data were pooled. Each sample used for statistical analysis consisted of data from all observations in the studied season.

## RESULTS

During the 3-year study, 5,807 specimens representing 84 species of Carabidae were caught (Table 1). Before the small retention program was implemented, 1,377 specimens belonging to 52 species were caught. In 2012, the number of species captured at the study site I did not change, although the number of carabid beetles caught (927 specimens) decreased. In the final year of the study, the number of species rose to 61, and their abundance increased to 1,493 specimens. At study site II, which was a replication, 965 specimens representing 57 species were caught, and the number of carabid individuals caught in 2013 increased to 1,045, while the number of species fell to 43 (Table 1). After the implementation of the water retention, the occurrence of new, hygrophilous species (*Patrobus atrorufus* Strøm, *Bembidion mannerheimii* C. R. Sahlberg) and an increase in the share of rare, disappearing or threatened species (*Oodes helopioides* Fabricius, *Carabus convexus* Fabricius, *Carabus marginalis* Fabricius) was observed. Moreover, following the implementation of the water retention program, the presence of a rare species, such as *Philorhizus sigma* P. ROSSI, was noted (Table 1). Despite the rise in the percentage of stenotopic species the hydro-technical works had been completed, the extremely endangered species *Epaphius rivularis* SCHRANK disappeared (Table 1).

The dominant species in both study sites and in all years of the study was the forest mesophilous beetle *Pterostichus niger* SCHALLER (2009: 22.15%, 2012 (I): 31.28%, 2012 (II): 32.75%, 2013 (I): 28.80%, 2013 (II): 32.34%). In the first year, another numerous species was the forest xerophilous *Calathus erratus* C.R. SAHLBERG (17.21%), which was not detected after the water retention project had been implemented. In 2012 and 2013, an increase occurred in the share of hygrophilous species classified as small predators of

**Table 1 Species composition, number of individuals, and structure of dominance in the Carabidae assemblages dwelling in the analyzed study sites located in the Dąbrówka Forest.**

| Study site | | 2009 | | 2012 | | | | 2013 | | | |
|---|---|---|---|---|---|---|---|---|---|---|---|
| | | I | | I | | II | | I | | II | |
| Species | Ecological description | Number | Dominance (%) | Number | Dominance (%) | Number | Dominance (%) | Number | Dominance (%) | Number | Dominance (%) |
| Pterostichus niger (SCHALLER, 1783) | F, M, Lz, Ab | 305 | 22.15 | 290 | 31.28 | 316 | 32.75 | 430 | 28.80 | 338 | 32.34 |
| Calathus erratus (C.R. SAHLBERG, 1827) | F, Xe, Sz, Ab | 237 | 17.21 | 0 | 0.00 | 0 | 0.00 | 0 | 0.00 | 0 | 0.00 |
| Harpalus tardus (PANZER, 1796) | Oa, Xe, Hz, Sb | 78 | 5.66 | 16 | 1.73 | 20 | 2.07 | 22 | 1.47 | 7 | 0.67 |
| Poecilus versicolor (STURM, 1824) | Oa, M, Sz, Sb | 96 | 6.97 | 39 | 4.21 | 52 | 5.39 | 50 | 3.35 | 41 | 3.92 |
| Carabus marginalis FABRICIUS, 1794 | F, M, Lz, Sb | 63 | 4.58 | 69 | 7.44 | 16 | 1.66 | 83 | 5.56 | 22 | 2.11 |
| Dyschirius globosus (HERBST, 1784) | Oa, H, Sz, Sb | 84 | 6.10 | 128 | 13.81 | 115 | 11.92 | 97 | 6.50 | 56 | 5.36 |
| Harpalus rufipes (DE GEER, 1774) | Oa, M, Hz, Ab | 28 | 2.03 | 7 | 0.76 | 2 | 0.21 | 7 | 0.47 | 1 | 0.10 |
| Epaphius secalis (PAYKULL, 1790) | Oa, M, Hz, Ab | 125 | 9.08 | 24 | 2.59 | 17 | 1.76 | 109 | 7.30 | 51 | 4.88 |
| Amara aenea (DE GEER, 1774) | Oa, Xe, Hz, Sb | 7 | 0.51 | 2 | 0.22 | 1 | 0.10 | 2 | 0.13 | 0 | 0.00 |
| Harpalus rubripes (DUFTSCHMID, 1812) | Oa, Xe, Hz, Sb | 5 | 0.36 | 4 | 0.43 | 1 | 0.10 | 3 | 0.20 | 2 | 0.19 |
| Amara lunicollis SCHIØDTE, 1837 | Oa, M, Hz, Ab | 26 | 1.89 | 5 | 0.54 | 7 | 0.73 | 10 | 0.67 | 5 | 0.48 |
| Amara communis (PANZER, 1797) | Oa, M, Hz, Sb | 18 | 1.31 | 12 | 1.29 | 15 | 1.55 | 40 | 2.68 | 16 | 1.53 |
| Bembidion gilvipes (STURM, 1825) | Oa, H, Sz, Sb | 78 | 5.66 | 39 | 4.21 | 74 | 7.67 | 51 | 3.42 | 32 | 3.06 |
| Amara convexior STEPHENS, 1828 | Oa, M, Hz, Sb | 2 | 0.15 | 6 | 0.65 | 6 | 0.62 | 24 | 1.61 | 15 | 1.44 |
| Poecilus lepidus (LESKE, 1785) | Oa, Xe, Sz, Sb | 10 | 0.73 | 3 | 0.32 | 0 | 0.00 | 4 | 0.27 | 0 | 0.00 |
| Amara bifrons (GYLLENHAL, 1810) | Oa, Xe, Hz, Ab | 14 | 1.02 | 0 | 0.00 | 0 | 0.00 | 1 | 0.07 | 0 | 0.00 |
| Pterostichus diligens (STURM, 1824) | Pb, H, Sz, Sb | 34 | 2.47 | 40 | 4.31 | 19 | 1.97 | 53 | 3.55 | 40 | 3.83 |
| Agonum fuliginosum (PANZER, 1809) | Eu, H, Sz, Sb | 32 | 2.32 | 61 | 6.58 | 28 | 2.90 | 112 | 7.50 | 63 | 6.03 |
| Pterostichus strenuus (PANZER, 1796) | F, H, Sz, Sb | 14 | 1.02 | 7 | 0.76 | 11 | 1.14 | 39 | 2.61 | 28 | 2.68 |
| Calathus melanocephalus (LINNAEUS, 1758) | Oa, M, Sz, Ab | 5 | 0.36 | 1 | 0.11 | 2 | 0.21 | 1 | 0.07 | 0 | 0.00 |
| Harpalus latus (LINNAEUS, 1758) | Eu, M, Hz, Ab | 7 | 0.51 | 2 | 0.22 | 6 | 0.62 | 5 | 0.33 | 3 | 0.29 |
| Clivina fossor (LINNAEUS, 1758) | Oa, M, Sz, Sb | 15 | 1.09 | 12 | 1.29 | 8 | 0.83 | 5 | 0.33 | 7 | 0.67 |
| Carabus nemoralis O.F. MÜLLER, 1764 | Oa, M, Sz, Sb | 7 | 0.51 | 14 | 1.51 | 24 | 2.49 | 9 | 0.60 | 23 | 2.20 |
| Harpalus smaragdinus (DUFTSCHMID, 1812) | Oa, Xe, Hz, Sb | 1 | 0.07 | 0 | 0.00 | 0 | 0.00 | 0 | 0.00 | 0 | 0.00 |
| Amara spreta DEJEAN, 1831 | Oa, Xe, Hz, Sb | 0 | 0.00 | 1 | 0.11 | 0 | 0.00 | 0 | 0.00 | 0 | 0.00 |
| Bradycellus harpalinus (AUDINET-SERVILLE, 1821) | Oa, Xe, Hz, Sb | 0 | 0.00 | 1 | 0.11 | 0 | 0.00 | 5 | 0.33 | 1 | 0.10 |
| Syntomus truncatellus (LINNE, 1761) | Oa, Xe, Sz, Sb | 6 | 0.44 | 1 | 0.11 | 3 | 0.31 | 3 | 0.20 | 0 | 0.00 |
| Synuchus vivalis (ILLIGER, 1798) | Oa, Xe, Sz, Ab | 5 | 0.36 | 0 | 0.00 | 2 | 0.21 | 3 | 0.20 | 1 | 0.10 |
| Carabus hortensis LINNAEUS, 1758 | F, M, Lz, Ab | 10 | 0.73 | 20 | 2.16 | 40 | 4.15 | 43 | 2.88 | 39 | 3.73 |
| Harpalus affinis (SCHRANK, 1781) | Eu, Xe, Hz, Sb | 4 | 0.29 | 0 | 0.00 | 1 | 0.10 | 0 | 0.00 | 1 | 0.10 |

| Study site | | 2009 | | 2012 | | | | 2013 | | | |
|---|---|---|---|---|---|---|---|---|---|---|---|
| | | I | | I | | II | | I | | II | |
| Species | Ecological description | Number | Dominance (%) | Number | Dominance (%) | Number | Dominance (%) | Number | Dominance (%) | Number | Dominance (%) |
| Epaphius rivularis (SCHRANK, 1781) | Pb, H, Sz, Ab | 15 | 1.09 | 0 | 0.00 | 0 | 0.00 | 0 | 0.00 | 0 | 0.00 |
| Amara tibialis (PAYKULL, 1798) | Oa, Xe, Hz, Sb | 0 | 0.00 | 1 | 0.11 | 0 | 0.00 | 3 | 0.20 | 0 | 0.00 |
| Bradycellus csikii LACZO, 1912 | Oa, M, Hz, Sb | 0 | 0.00 | 0 | 0.00 | 0 | 0.00 | 1 | 0.07 | 0 | 0.00 |
| Leistus terminatus PANZER, 1793 | Pb, H, Sz, Ab | 1 | 0.07 | 9 | 0.97 | 10 | 1.04 | 21 | 1.41 | 8 | 0.77 |
| Loricera pilicornis (FABRICIUS, 1775) | Pb, H, Sz, Sb | 1 | 0.07 | 0 | 0.00 | 0 | 0.00 | 2 | 0.13 | 0 | 0.00 |
| Poecilus cupreus (LINNAEUS, 1758) | Oa, M, Sz, Sb | 1 | 0.07 | 0 | 0.00 | 0 | 0.00 | 0 | 0.00 | 0 | 0.00 |
| Pterostichus vernalis (PANZER, 1796) | Oa, M, Sz, Sb | 4 | 0.29 | 3 | 0.32 | 3 | 0.31 | 4 | 0.27 | 3 | 0.29 |
| Amara curta DEJEAN, 1828 | Oa, Xe, Hz, Sb | 6 | 0.44 | 13 | 1.40 | 8 | 0.83 | 10 | 0.67 | 3 | 0.29 |
| Bembidion mannerheimii (C. R. SAHLBERG, 1827) | Eu, H, Sz, Sb | 0 | 0.00 | 0 | 0.00 | 2 | 0.21 | 4 | 0.27 | 0 | 0.00 |
| Patrobus atrorufus (STRØM, 1768) | Pb, H, Sz, Ab | 0 | 0.00 | 0 | 0.00 | 2 | 0.21 | 6 | 0.40 | 6 | 0.57 |
| Amara brunnea (GYLLENHAL, 1810) | F, M, Hz, Ab | 1 | 0.07 | 0 | 0.00 | 0 | 0.00 | 1 | 0.07 | 0 | 0.00 |
| Calathus fuscipes GOEZE, 1777 | Oa, M, Sz, Ab | 1 | 0.07 | 0 | 0.00 | 0 | 0.00 | 0 | 0.00 | 0 | 0.00 |
| Carabus convexus FABRICIUS, 1775 | F, Xe, Lz, Sb | 0 | 0.00 | 7 | 0.76 | 3 | 0.31 | 16 | 1.07 | 10 | 0.96 |
| Harpalus griseus (PANZER, 1796) | Oa, Xe, Hz, Ab | 0 | 0.00 | 0 | 0.00 | 1 | 0.10 | 0 | 0.00 | 0 | 0.00 |
| Microlestes minutulus (GOEZE, 1777) | Oa, Xe, Hz, Ab | 0 | 0.00 | 1 | 0.11 | 0 | 0.00 | 0 | 0.00 | 0 | 0.00 |
| Oxypselaphus obscurus (HERBST, 1784) | F, H, Sz, Sb | 1 | 0.07 | 4 | 0.43 | 1 | 0.10 | 4 | 0.27 | 5 | 0.48 |
| Amara equestris (DUFTSCHMID, 1812) | Oa, Xe, Hz, Ab | 1 | 0.07 | 0 | 0.00 | 0 | 0.00 | 2 | 0.13 | 0 | 0.00 |
| Amara familiaris (DUFTSCHMID, 1812) | Eu, M, Hz, Sb | 1 | 0.07 | 0 | 0.00 | 3 | 0.31 | 0 | 0.00 | 0 | 0.00 |
| Bembidion guttula (FABRICIUS, 1792) | Oa, H, Sz, Sb | 0 | 0.00 | 0 | 0.00 | 1 | 0.10 | 1 | 0.07 | 0 | 0.00 |
| Carabus granulatus LINNAEUS, 1758 | Pb, H, Lz, Sb | 3 | 0.22 | 18 | 1.94 | 11 | 1.14 | 37 | 2.48 | 28 | 2.68 |
| Pterostichus minor (GYLLENHAL, 1827) | Pb, H, Sz, Sb | 4 | 0.29 | 3 | 0.32 | 1 | 0.10 | 16 | 1.07 | 4 | 0.38 |
| Acupalpus exiguus DEJEAN, 1829 | Pb, H, Hz, Sb | 1 | 0.07 | 1 | 0.11 | 0 | 0.00 | 1 | 0.07 | 0 | 0.00 |
| Amara similata (GYLLENHAL, 1810) | Oa, M, Hz, Sb | 1 | 0.07 | 3 | 0.32 | 0 | 0.00 | 1 | 0.07 | 0 | 0.00 |
| Calathus ambiguus (PAYKULL, 1790) | Oa, Xe, Sz, Ab | 1 | 0.07 | 0 | 0.00 | 0 | 0.00 | 0 | 0.00 | 0 | 0.00 |
| Carabus cancellatus ILLIGER, 1798 | Oa, Xe, Sz, Ab | 0 | 0.00 | 2 | 0.22 | 1 | 0.10 | 0 | 0.00 | 0 | 0.00 |
| Carabus glabratus PAYKULL, 1790 | F, M, Lz, Ab | 0 | 0.00 | 2 | 0.22 | 3 | 0.31 | 7 | 0.47 | 14 | 1.34 |
| Harpalus luteicornis (DUFTSCHMID, 1812) | Oa, M, Hz, Sb | 7 | 0.51 | 0 | 0.00 | 1 | 0.10 | 0 | 0.00 | 0 | 0.00 |
| Leistus ferrugineus LINNAEUS, 1758 | F, M, Sz, Ab | 1 | 0.07 | 0 | 0.00 | 2 | 0.21 | 4 | 0.27 | 2 | 0.19 |
| Notiophilus palustris (DUFTSCHMID, 1812) | Oa, M, Sz, Sb | 1 | 0.07 | 1 | 0.11 | 5 | 0.52 | 5 | 0.33 | 2 | 0.19 |
| Oodes helopioides (FABRICIUS, 1792) | Pb, H, Sz, Sb | 3 | 0.22 | 19 | 2.05 | 5 | 0.52 | 29 | 1.94 | 22 | 2.11 |
| Pterostichus nigrita (PAYKULL, 1790) | F, H, Sz, Sb | 1 | 0.07 | 1 | 0.11 | 1 | 0.10 | 3 | 0.20 | 2 | 0.19 |

(Continued)

| Study site | | 2009 | | 2012 | | | | 2013 | | | |
|---|---|---|---|---|---|---|---|---|---|---|---|
| | | I | | I | | II | | I | | II | |
| Species | Ecological description | Number | Dominance (%) | Number | Dominance (%) | Number | Dominance (%) | Number | Dominance (%) | Number | Dominance (%) |
| *Pterostichus oblongopunctatus* (FABRICIUS, 1787) | F, M, Sz, Sb | 0 | 0.00 | 17 | 1.83 | 88 | 9.12 | 74 | 4.96 | 104 | 9.95 |
| *Stenolophus mixtus* (HERBST, 1784) | Oa, H, Sz, Sb | 0 | 0.00 | 0 | 0.00 | 0 | 0.00 | 1 | 0.07 | 1 | 0.10 |
| *Agonum sexpunctatum* (LINNAEUS, 1758) | Oa, H, Sz, Sb | 1 | 0.07 | 3 | 0.32 | 2 | 0.21 | 6 | 0.40 | 18 | 1.72 |
| *Agonum viduum* (PANZER, 1796) | Pb, H, Sz, Sb | 0 | 0.00 | 0 | 0.00 | 0 | 0.00 | 1 | 0.07 | 0 | 0.00 |
| *Amara plebeja* (GYLLENHAL, 1810) | Oa, H, Hz, Sb | 0 | 0.00 | 0 | 0.00 | 1 | 0.10 | 1 | 0.07 | 0 | 0.00 |
| *Anisodactylus binotatus* (FABRICIUS, 1787) | Oa, H, Hz, Sb | 2 | 0.15 | 1 | 0.11 | 4 | 0.41 | 3 | 0.20 | 1 | 0.10 |
| *Badister bullatus* (SCHRANK, 1798) | F, M, Sz, Sb | 0 | 0.00 | 1 | 0.11 | 2 | 0.21 | 0 | 0.00 | 0 | 0.00 |
| *Badister lacertosus* STURM, 1815 | F, M, Sz, Sb | 0 | 0.00 | 0 | 0.00 | 1 | 0.10 | 0 | 0.00 | 0 | 0.00 |
| *Badister sodalis* (DUFTSCHMID, 1812) | Pb, H, Sz, Sb | 0 | 0.00 | 1 | 0.11 | 0 | 0.00 | 2 | 0.13 | 0 | 0.00 |
| *Bembidion doris* (PANZER, 1796) | Pb, H, Sz, Sb | 0 | 0.00 | 0 | 0.00 | 0 | 0.00 | 1 | 0.07 | 0 | 0.00 |
| *Metallina properans* (STEPHENS, 1828) | Oa, M, Sz, Sb | 0 | 0.00 | 1 | 0.11 | 1 | 0.10 | 0 | 0.00 | 0 | 0.00 |
| *Bembidion quadrimaculatum* (LINNAEUS, 1760) | Oa, M, Sz, Sb | 0 | 0.00 | 0 | 0.00 | 0 | 0.00 | 1 | 0.07 | 0 | 0.00 |
| *Calathus micropterus* (DUFTSCHMID, 1812) | F, M, Sz, Ab | 2 | 0.15 | 5 | 0.54 | 10 | 1.04 | 10 | 0.67 | 17 | 1.63 |
| *Carabus arvensis* HERBST, 1784 | F, Xe, Lz, Sb | 0 | 0.00 | 2 | 0.22 | 1 | 0.10 | 2 | 0.13 | 1 | 0.10 |
| *Chlaenius nigricornis* (FABRICIUS, 1787) | Oa, H, Sz, Sb | 0 | 0.00 | 1 | 0.11 | 1 | 0.10 | 0 | 0.00 | 0 | 0.00 |
| *Cicindela hybrida* LINNAEUS, 1758 | Oa, Xe, Lz, Sb | 0 | 0.00 | 0 | 0.00 | 0 | 0.00 | 1 | 0.07 | 0 | 0.00 |
| *Harpalus laevipes* ZETTERSTEDT 1828 | F, M, Hz, Sb | 0 | 0.00 | 0 | 0.00 | 1 | 0.10 | 1 | 0.07 | 1 | 0.10 |
| *Lebia chlorocephala* (J. J. HOFFMANN, 1803) | Oa, M, Sz, Sb | 0 | 0.00 | 2 | 0.22 | 0 | 0.00 | 0 | 0.00 | 0 | 0.00 |
| *Limodromus assimilis* PAYKULL, 1790 | F, H, Sz, Sb | 0 | 0.00 | 1 | 0.11 | 0 | 0.00 | 0 | 0.00 | 0 | 0.00 |
| *Nebria brevicollis* (FABRICIUS, 1792) | Eu, M, Sz, Ab | 0 | 0.00 | 0 | 0.00 | 0 | 0.00 | 0 | 0.00 | 1 | 0.10 |
| *Notiophilus biguttatus* (FABRICIUS, 1779) | F, M, Sz, Sb | 0 | 0.00 | 0 | 0.00 | 1 | 0.10 | 0 | 0.00 | 0 | 0.00 |
| *Philorhizus sigma* (P. ROSSI, 1790) | Oa, Xe, Sz, Sb | 0 | 0.00 | 0 | 0.00 | 1 | 0.10 | 0 | 0.00 | 0 | 0.00 |
| *Pterostichus anthracinus* (ILLIGER, 1798) | Pb, H, Sz, Ab | 0 | 0.00 | 0 | 0.00 | 1 | 0.10 | 0 | 0.00 | 0 | 0.00 |
| Number of species | | 52 | | 52 | | 57 | | 61 | | 43 | |
| | | 84 | | | | | | | | | |
| Individuals-total | | 1,377 | 100.00 | 927 | 100.00 | 965 | 100.00 | 1,493 | 100.00 | 1,045 | 100.00 |
| | | 5,807 | | | | | | | | | |

**Table 2 Results of the generalized linear model (GLM) of the analyzed transects and years when the observations were conducted, for the basic parameters describing the biodiversity of Carabidae assemblages in the analyzed area.**

| Species | df | Wald's statistic | p |
|---|---|---|---|
| Year | 4 | 90.99 | 0.00 |
| Transect | 2 | 18.44 | 0.00 |
| Year * transect | 8 | 67.58 | 0.00 |
| Individuals | | | |
| Year | 4 | 287.81 | 0.00 |
| Transect | 2 | 51.55 | 0.00 |
| Year * transect | 8 | 216.91 | 0.00 |
| Shannon H′ | | | |
| Year | 4 | 25.14 | 0.00 |
| Transect | 2 | 3.42 | 0.18 |
| Year * transect | 8 | 23.12 | 0.00 |
| MIB | | | |
| Year | 4 | 40.90 | 0.00 |
| Transect | 2 | 879.70 | 0.00 |
| Year * transect | 8 | 12.20 | 0.14 |

**Table 3 Number of species, individuals and diversity index of carabids assemblages in the analyzed areas of Dąbrówka peat bog.**

| Specification | 2009 | | | 2012_1 | | | 2012_2 | | | 2013_1 | | | 2013_2 | | | Total |
|---|---|---|---|---|---|---|---|---|---|---|---|---|---|---|---|---|
| | A | B | C | A | B | C | A | B | C | A | B | C | A | B | C | |
| Number of individuals | 680 | 480 | 217 | 360 | 322 | 245 | 314 | 326 | 325 | 497 | 486 | 510 | 316 | 316 | 413 | 5,807 |
| Number of species | 41 | 36 | 31 | 30 | 40 | 29 | 40 | 34 | 31 | 43 | 45 | 35 | 31 | 35 | 26 | 84 |
| Shannon H′ diversity | 1.25 | 0.8 | 0.4 | 0.44 | 0.59 | 0.48 | 0.5 | 0.56 | 0.55 | 0.81 | 0.85 | 0.96 | 0.67 | 0.68 | 0.63 | – |

the spring type of development, they may be eurytopic or they may be associated with open areas (*Dyschirius globosus* HERBST, *Agonum fuliginosum* PANZER, *Agonum sexpunctatum* LINNAEUS) or connected with forests as their habitat (*Pterostichus strenuus* PANZER, *Oxypselaphus obscurus* HERBST). In addition, following the water retention, there was a rise in the abundance of peatland hygrophilous beetles (*Carabus granulatus* LINNAEUS, *Oodes helopioides* FABRICIUS, *Pterostichus diligens* STURM, *Leistus terminatus* PANZER, *Patrobus atrorufus* STRØM) (Table 1).

The results obtained from the generalized linear model (GLM) showed a statistically significant impact of the variable factors (transect, year) and their interaction with respect to the presence of epigeic carabid beetles (Table 2). The GLM model was chosen because the data had a unimodal distribution. The transects chosen in the study differed from each other in the quantities of individuals and number of species of ground beetles caught (Table 3).
The highest values concerning the number of species and number of individuals prior to the hydro-technical modifications were noted in transect A (peat meadow), while the lowest ones were in transect C (forest) (Figs. 2A and 2B). This is supported by the biological diversity assessment, which additionally demonstrated the most significant contrast during the 3-year research between transects A (1.25) and C (0.4) (Table 3). After the execution of the small water retention project, there were no quantitative or qualitative dependences confirmed proportional to the distance of the determined transects from the retention reservoirs (Figs. 2A and 2B). In 2012, the species richest transect was transect B (forest-meadow ecotone), while the fewest species were caught in transect C (forest). In 2013, the final year of our observations, the above values in individual transects were approximately the same. The MIB analysis showed that the transect localized in a forest habitat (C) was characterized by the highest values in each year of the study (Fig. 2C).

Species accumulation curves for individual transects in the 3 years of the study confirmed that the sampling effort was adequate (Fig. 3). Only the rarefaction curve from transect A, located closest to the water reservoirs, in the year of transformation (2012) did not reach an asymptote. This was probably due to unstable conditions leading to fauna migration, which is reflected in the number of singletons (over 50%).

Graphical and statistical analysis of the interpretation of differences in particular transects was presented with the aid of the non-metric multidimensional scaling (NMDS) analysis. The analysis showed that the Carabidae assemblages inhabiting the analyzed transects, before and after the implementation of the small retention program, were significantly different between one another (ANOSIM; A: $R = 0.30$, B: $R = 0.26$, C: $R = 0.32$; $p < 0.01$). The curves in the graphs illustrating the similarity between the captured Carabidae assemblages in the individual years of entomological observations (Stress $A = 0.22$, $B = 0.16$, $C = 0.14$) demonstrated that the statistically most significant difference was in transect A (Figs. 4A–4C).

RDA analysis showing the relationships between ecological groups of ground beetles and environmental variables (year, transect) showed that the first ordination axis, determining 82.4% of diversity, is positively correlated with the occurrence of large forest zoophages, whose occurrence determines the growth of MIB. This condition is characteristic of stable habitats, which is typical for forest areas (transect C). An inverse correlation was found for transects A and B. Transect A was correlated with the presence of hygrophilous species associated with peat bogs, while transect B showed a strong correlation with the occurrence of small zoophages belonging to eurytopic and open area species. RDA analysis also showed that the research years were correlated mainly with the second ordination axis, describing almost 10% of the variation. The highlighted factor-2009-is correlated with the occurrence of xerophilic hemizoophages with an autumn type of development. The vector describing 2012 showed an inverse correlation, which indicates large changes in ground beetle assemblages as a result of earth transformations. On the other hand, the vector describing 2013 revealed a tendency corresponding to the state of the carabid assemblages in 2009, which indicates the tendency of the structure of carabid

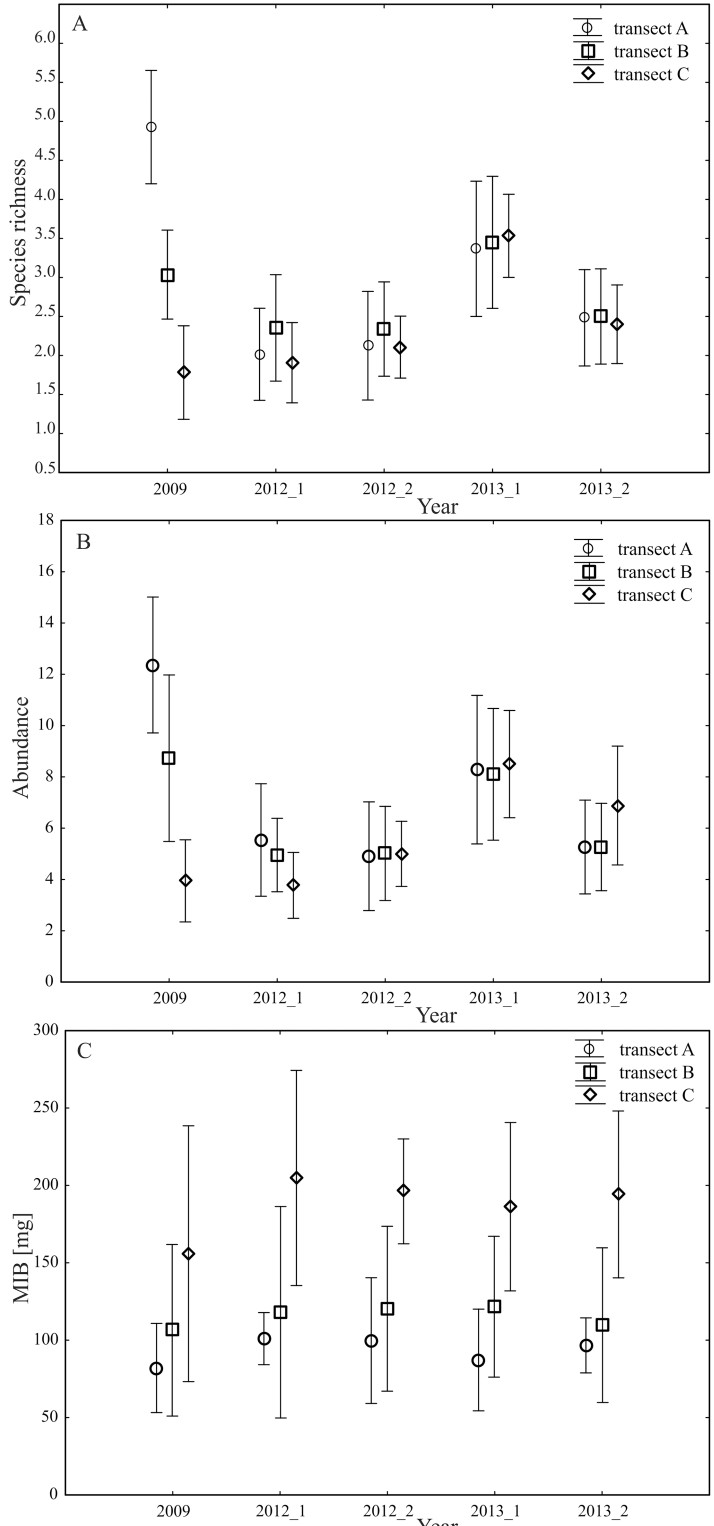

**Figure 2** (A–C) Mean values of species richness (A), abundance (B) and the MIB (C) in the analyzed transects and years of study, vertical lines with average mean SE.

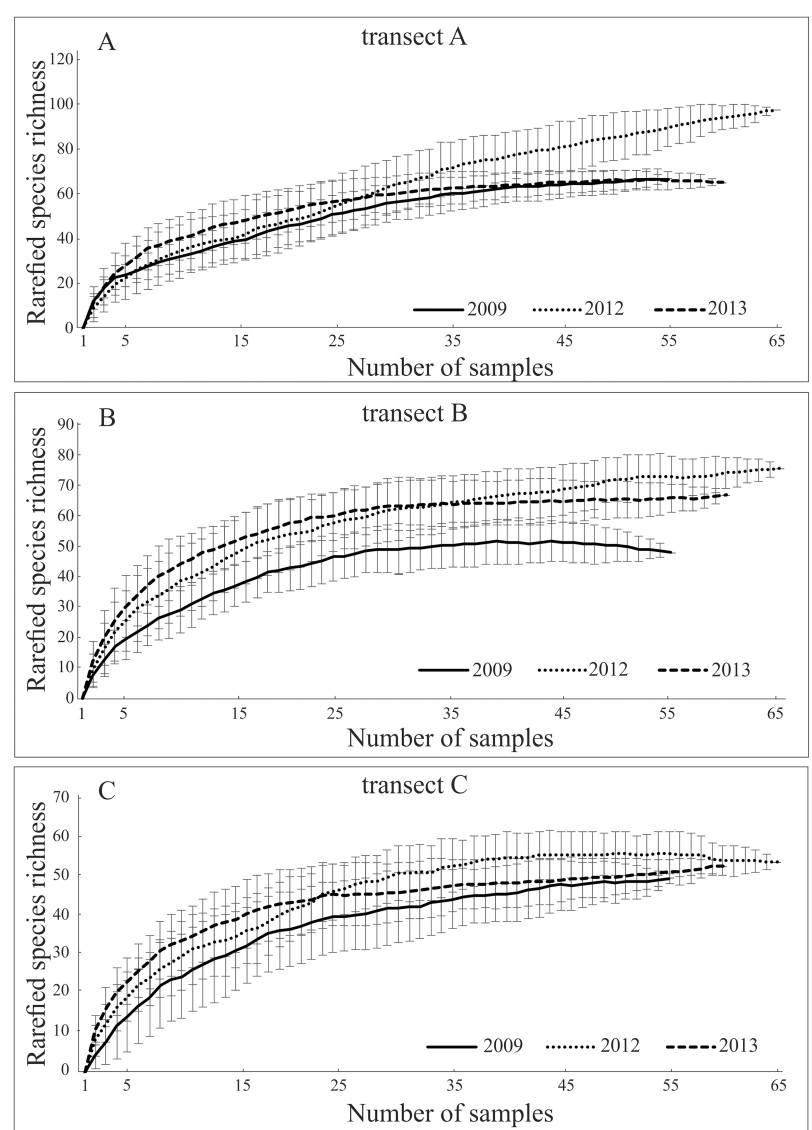

**Figure 3 (A–C) Expected number of carabid species caught in the studied sites using the Jackknife estimator (±SD) of species richness (Species accumulation curves for carabid beetles assemblages sampled in three transects in years of study, using the Jackknife).**

assemblages to return to the state before implementation of the water retention program (Fig. 5).

## DISCUSSION

This study, conducted in the Dąbrówka Forest Subdistrict, showed that the number of individuals and species diversity of the captured carabid beetles was at a high level in comparison with results achieved on peatlands in Belarus, Germany and Slovakia (*Aleksandrowicz, 2002*; *Buchholz, Hannig & Schirmel, 2009*; *Igondová & Majzlan, 2015*). In our study, we observed over 29% of all carabid species present in the Masurian Lake District, which may suggest that the analyzed area plays a significant role in the

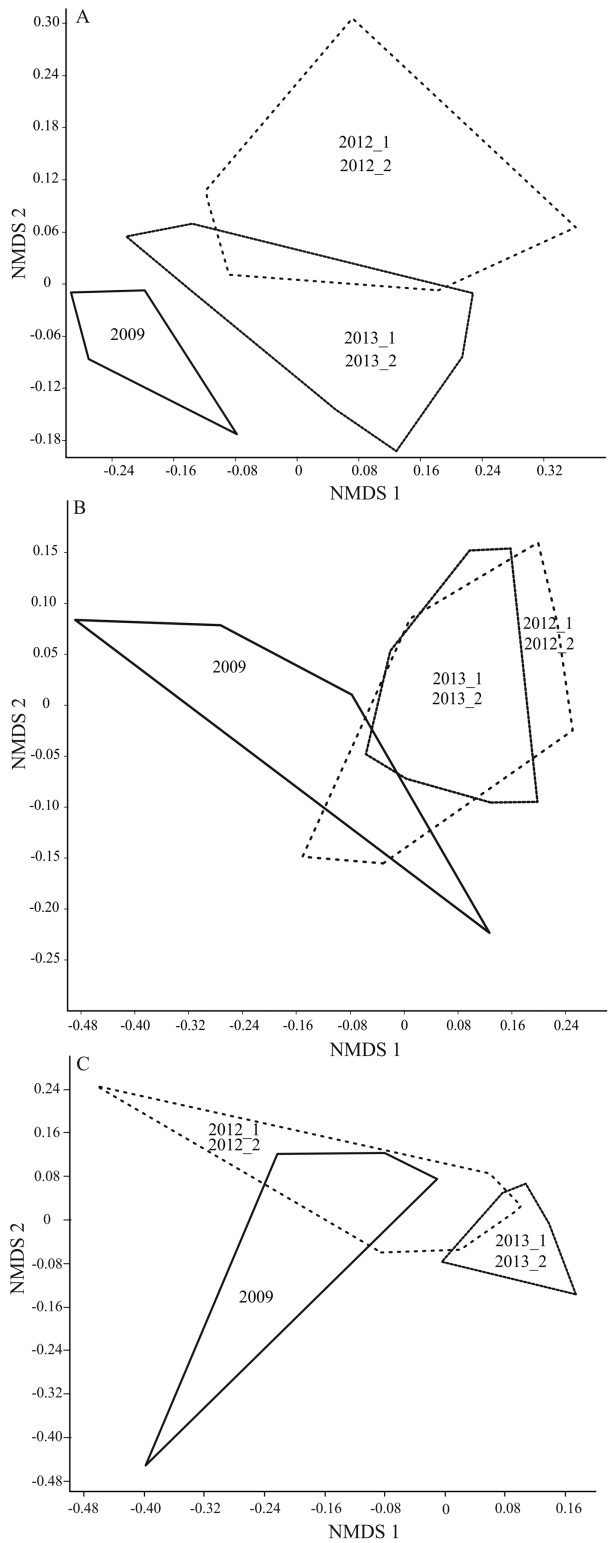

**Figure 4 (A–C) Diagram of the non-metric multidimensional scaling (NMDS) conducted based on the Morisita's similarity measure for the transects designated in the experiment (A–C).**

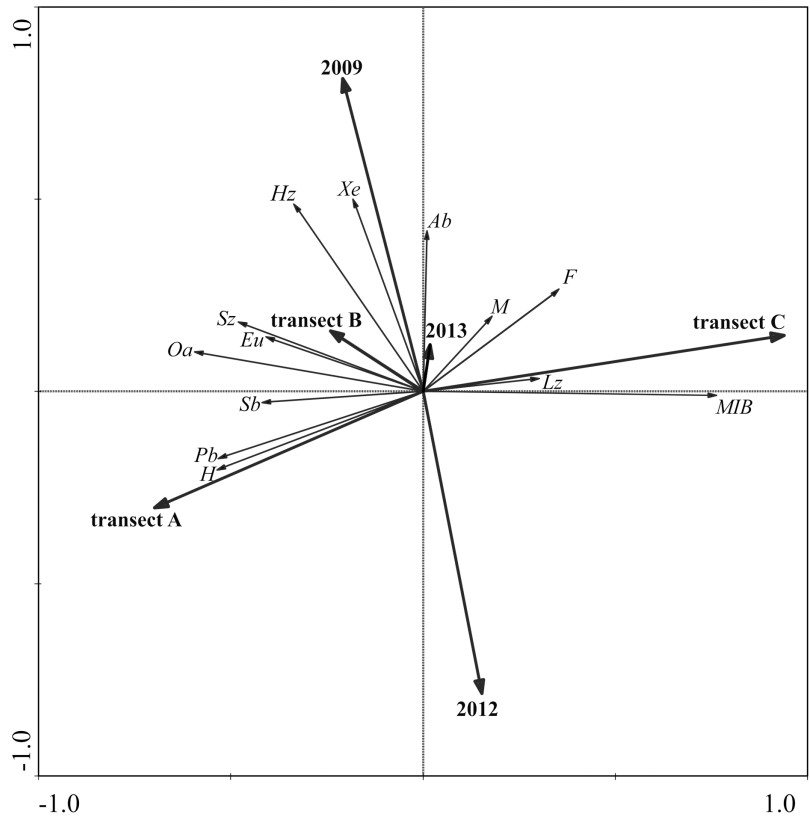

**Figure 5 Diagram of the RDA analysis demonstrating the relationships between the analyzed environmental variables and Carabidae species in the studied sites.** The abbreviation used on the graph are described in the "Materials and Methods" section.

preservation of biological diversity (*Aleksandrowicz, Gawroński & Browarski, 2003*; *Pacuk & Regulska, 2014*). Similar conclusions can be drawn from analysis the quantitative and qualitative composition of carabid beetle assemblages from peatlands located in the north-east of Poland (*Nietupski et al., 2008*; *Nietupski, Ciepielewska & Kosewska, 2008*; *Nietupski et al., 2010*; *Nietupski, Kosewska & Lemkowska, 2015*). Because of the poor knowledge of entomofauna inhabiting areas subjected to water retention programs, *Homburg et al. (2014)* suggest that it is necessary to gain more insight into this research question. Our studies conducted in the Olsztyn Forest District have demonstrated that the impact of hydro-technical works directly affect the typical habitat conditions and the composition of caught entomofauna. *Kiryluk (2012)* and *Abell et al. (2019)* confirm that when moisture conditions deteriorate, the diversity of flora decreases, which causes impoverishment of fauna. Hence, this is another parameter which directly affects the species composition and the abundance of captured insects, by being a factor that decides about their ability to survive, thereby differentiating the quality of insect assemblages (*Thiele, 1977*; *Nietupski, Kosewska & Ciepielewska, 2006*). *Jaworski & Hilszczański (2013)* emphasize that temperature and humidity have both direct and indirect effects on all aspects of the life of both the larvae and imagines of insects.

Out of the three years of observations, the entomological material gathered in the first year after the hydro-technical modifications was the least numerous, but this may have been caused by stronger anthropopressure on the analyzed ecosystems. Similar conclusions are drawn by *Skłodowski & Mądrzejewska (2008)* and by *Rosenzweig (1995)*, who claim that environmental disturbances have a considerable influence, leading to the reduced biodiversity of local ecosystems. Moreover, that year was characterized by relatively high atmospheric precipitation (700 mm), which could also have had an influence on the presence of carabid beetles (*Central Statistical Office in Poland, 2013*). In turn, the increase in the diversity of carabids in the last year of the study (2013) may be indicative of the natural conditions slowly normalizing after the implementation of changes in the water relations, whose main goal was to improve the moisture conditions. These results are in accord with those reported by *Jędryczkowski & Kupryjanowicz (2005)*, who emphasize that species diversity grows proportionally to the increase in the area's moisture content. The effect of soil moisture in forest habitats on beetle assemblages was also demonstrated by *Kagawa & Maeto (2014)*.

Changes in the habitat induced by the implementation of the small water retention program in the Dąbrówka Forest Subdistrict also affected the distribution of the carabid beetles in the three research transects. Before the earthworks, the diversity of carabid species decreased directly proportionally to the increasing distance of the set transects from the open area of the peatland meadow (Table 3). The earthworks caused certain disturbances in the habitat, associated, inter alia, with increased soil moisture, which caused changes in the spatial structure of carabid assemblages. Fluctuations in the natural environment due to the retention works created stress conditions, which were most distinctly evidenced during the 3-year period in transect A. Transect A, located closest to the retention reservoirs, responded the most to the induced changes, which is reflected in the graphical shift of the polygon describing the similarity of the Carabidae assemblages in this transect in the multidimensional spatial scaling approach (NMDS) (Figs. 4A–4C). Following the hydro-technical modifications, the species-richest study site (except study site 2012 II) was transect B (Table 3). This transect, by being localized at the edge of two different habitats, could have been conducive to a rise in biodiversity (*Yu et al., 2007*; *Banul, Kosewska & Borkowski, 2018*). *Szyszko (2002)* states that transitional zones between two different habitats are an excellent foraging site, and subsequently a good place for the development of carabid beetles. Additionally, the above transect in our study was an area subjected to a lesser influence of environmental disturbances, which may also have contributed to higher species diversity. The lowest diversity was noted in the forest zone (transect C) (Table 3). Forest habitats are characterized by relatively low diversity, but these assemblages are stable and quite resistant to environmental changes. This opinion has been verified by the observations made in the forest ecosystem located in the Dąbrówka Forest Subdistrict (C) (Figs. 4A–4C). The role of the stable woodland ecosystem in shaping species diversity and abundance of insects has been highlighted by *Sushko (2014)*, who carried out entomological studies in Lithuania. Sushko showed a greater diversity of carabid fauna in a birch forest than in the adjacent peatland. Furthermore, he detected similar species

diversity of carabid beetles in a pine forest and on peatland, but determined that the number of beetles in the pine forest was twice as high as on the peatland (*Sushko, 2007*). *Dapkus & Tamutis (2008)* reported that the number of species on peatland and in a bordering pine forest was similar, but twice as many carabid specimens were detected in the forest than on the peatland.

The adaptability of carabid beetles in the course of evolution led to the emergence of individuals with changeable preferences in regard to habitat, moisture and trophy, or representing different development types (*Thiele, 1977*). Our analysis, in line with the division adopted for the sake of this research, showed the presence of representatives of all categories of the carabid fauna living in Poland, in terms of their preferred habitat, moisture conditions and trophy. This may be an argument in favor of the internal diversity of the analyzed habitats, and further indicating the high diversity of insects inhabiting this area. The fluctuations that had appeared shaped the structures of biocenoses and the created environmental factors gave rise to new, often valuable populations (*Skłodowski & Zdzioch, 2006*). *Czyżyk & Porter (2017)* undertook a study to assess the influence of small water reservoirs created in woodlands on the surrounding environment. The results they obtained showed the impact of the water bodies on the food base for Carabidae, as a result of which structural changes occurred in carabid assemblages. Additionally, analysis of transects located at different distances from the water reservoirs revealed the effect of the distance to these study sites on the quantitative and qualitative composition of epigeic carabid beetle assemblages. The differences observed between Carabidae populations in the years of our study, presented via NMDS analysis, only indicate the significance and general direction of changes, which were probably effected by habitat transformations caused by water- and earthworks. A more detailed analysis of changes occurring in carabid assemblages is provided by the RDA redundancy analysis (Fig. 5). The completed earthworks caused some changes in the habitat, including higher soil moisture. This resulted in a decrease in the share of xerophilic hemizoophages with autumn type of development and an increase in the share of zoophages with higher water requirements, the spring type of development inhabiting forests, open areas and peat bogs, or being eurytopes.

The character of changes in the structure of epigeic carabid assemblages induced by the land retention works, in general, should be perceived as positive, in agreement with the expectations of what condition this type of habitat ought to be in *Czyżyk & Porter (2017)*. *Zabrocka-Kostrubiec (2008)* concludes that measures implemented under the umbrella of small water retention programs play a significant and beneficial role in forest management practice and, in the long term, they assure the permanently sustainable development of woodland ecosystems included in such programs. However, the effects are not unidirectional, because water retention transformations also lead to the disappearance of certain habitats (*Bajkowski et al., 2000*), and in the case of peatland and marsh habitats, they can result in considerable ecological transformations (*Grzywna, 2010*). Thus, certain concerns can be raised by the fact that such rapid habitat-related changes may result in the disappearance of rare, stenotopic species, such as *E. rivularis* in our study. The question of whether this is a permanent outcome, or whether that species

will reappear in the habitat after some time, can be resolved only through further observations.

## CONCLUSIONS

Water as a scarce product is a current problem worldwide. For this reason, programs are being implemented whose main goal is to prevent water shortages. One such program is small water retention. When introducing such changes, however, we must be aware that retention works induce transformations of habitats, and thereby effect changes in the composition of valuable Carabidae fauna. The implementation of this program, on the one hand, causes a decrease in the number of carabid beetles and certain disturbances in the previous structure of their assemblages but, on the other hand, it enables the appearance of Carabidae fauna typical for this type of habitat, with specific habitat and moisture requirements, often valuable in the natural environment. It can, therefore, be concluded that small water retention projects have a strong effect on epigeic fauna, and the transformation mostly involves xerophilous species being replaced by hygrophilous species, with greater ecological adaptability (eurytopic species with the spring type of development). Another demonstrable change is the increase in the number of peat bog carabids, which indicates the direction of habitat-related changes induced by hygrotechnical works. In consequence, the area undergoes transformation and is again settled by hygrophilous organisms, which are otherwise often on the brink of extinction. However, only many further years of research and monitoring of retention areas could give an answer as to whether these processes are short-term and reversible or not.

## ACKNOWLEDGEMENTS

We would like to thank the Olsztyn Forest District for help with material collection.

### Funding

This work was supported by a research project of the University of Warmia and Mazury in Olsztyn (no. 20.610.015-110). The funders had no role in study design, data collection and analysis, decision to publish, or preparation of the manuscript.

### Grant Disclosures

The following grant information was disclosed by the authors:
Warmia and Mazury in Olsztyn: 20.610.015-110.

### Competing Interests

The authors declare that they have no competing interests.

### Author Contributions

- Emilia Ludwiczak performed the experiments, analyzed the data, prepared figures and/or tables, authored or reviewed drafts of the paper, and approved the final draft.

# PeerJ

- Mariusz Nietupski conceived and designed the experiments, performed the experiments, analyzed the data, prepared figures and/or tables, authored or reviewed drafts of the paper, and approved the final draft.
- Agnieszka Kosewska performed the experiments, analyzed the data, prepared figures and/or tables, authored or reviewed drafts of the paper, and approved the final draft.

## Data Availability

Raw measurements are available in the Supplemental Files.

## Supplemental Information

Supplemental information for this article can be found online at http://dx.doi.org/10.7717/peerj.9815#supplemental-information.

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
