# Peer review of "Ground beetles (Coleoptera; Carabidae) as an indicator of ongoing changes in forest habitats due to increased water retention"

_PeerJ, doi:10.7717/peerj.9815_

## Round 0.1 · original submission · Major Revisions

Dear Dr. Ludwiczak and colleagues:

Thanks for submitting your manuscript to PeerJ. I have now received three independent reviews of your work, and as you will see, the reviewers raised some major concerns about the research (and the manuscript). Despite this, these reviewers are optimistic about your work and the potential impact it will have on research studying environmental impacts on ground beetles. Thus, I encourage you to revise your manuscript, accordingly, taking into account all of the concerns raised by both reviewers.

Most importantly, two of the three reviewers were more optimistic, while one reviewer recommended rejection. This reviewer (2) is concerned about gross misuse of terminology, as well a failure to clearly state your research questions. Please address these issues.

The supplemental files seem to be incomplete and need refining.

Please take more care and precision in explaining the results.

There are many minor problems pointed out by the reviewers, and you will need to address all of these and expect a thorough review of your revised manuscript by these same reviewers.

I agree with the concerns of the reviewers, and thus feel that their suggestions should be adequately addressed before moving forward. Please note that Reviewer 1 kindly provided a marked-up version of your manuscript.

Therefore, I am recommending that you revise your manuscript, accordingly, taking into account all of the issues raised by the reviewers.

I look forward to seeing your revision, and thanks again for submitting your work to PeerJ.

Good luck with your revision,

-joe

Reviewer 1 ·

Basic reporting

Title
The title is clear and properly reflect the content.
Abstract and key words
The abstract is suitable and basically good.
Line 17 I think that the aim of your study is not "to try" but just "to assess". Please improve it.
Line 17-20 This sentence is definitely too long. Please divide it. It will make the goal more understandable and transparent.
Line 20-21 Sentence need the language improvements.
Line 24 Remove "observed". Please avoid redundant sentence structure or words.
Introduction
The introduction is comprehensive, but needs some corrections and explanations. Moreover in the title of the paper you use " ground beetle (Coleoptera; Carabidae) as an indicator...", please develop this thread in more detail in the introduction.
Line 52-54 Please do the language improvements of this sentence.
Line 60 I propose to add some reference to confirm this statement.
Line 61-65 Too long sentence. Please divide it.
Line 67 Remove "to try", it is not a scientific statement. Use just "evaluate..."
Discussion and Conclusions
Discussion is interesting and all-embracing.
References
Literature is well-chosen, but only three articles are from the last three years. Can you add more recent references? It will also emphasize that your research is current.
Table 2 There is "zone". Please clarify it and be consistent throughout the text.
Table 3. Please do not double Carabid/Carabidae in table caption.
Figure 2 Please remove the grid lines from the chart. Figure 2 a- axis signature should be "species richness", b- abundance. It should be the same as in the figure caption. The style of figure 2c differs slightly from a and b.
Figure 3 Please remove the grid lines from the chart. Please sign the axes on figure A, B and C.
Figure 4. Please sign the axes on figure A, B and C. Add the meaning of the abbreviations to the figure caption.

Experimental design

Material and Methods
Line 75-79 Once again, too long sentence.
Line 79 "Object" is not appropriate term. Maybe please use for example "study site"
Line 85-88 What does it mean "sphere". It's a strange term, rather rarely used. For me it looks like transect of five pitfall traps...or three habitats (A, B, C).
Line 94-95 Use English name in brackets.
Line 102 "and" should be remove, use comma.
Line 112 Here you use transect, and it is good expression. Use the same term throughout your work. I think the term sphere is misleading.
Line 121 "During the capture of beetles..." this is not good English expression. Please improve it.
Line 152 It should be GLM, not GML. Did you check the distribution of variables. You don't mention it in Data analysis.
Line 157 Why you used the Morisita's measure of similarity, not e.g. Bray-Curtis?
Line 161 I am not sure that PCA is appropriate analysis. Principal components analysis is a variable-reduction technique that shares many similarities to exploratory factor analysis. Its aim is to reduce a larger set of variables into a smaller set of 'artificial' variables, called 'principal components', which account for most of the variance in the original variables. In your case I propose to use RDA analysis (Redundancy analysis with Monte Carlo permutation test) because it is a method to extract and summarize the variation in a set of response variables that can be explained by a set of explanatory variables.
Line 167 "species classified as belonging to..." , please delete "classified as".
Line 178 Remove "once"
Line 194-198 Too long sentence
Line 219 What percentage of variance do the axes describe? As above I think the RDA analysis would be appropriate here not PCA.

Validity of the findings

no comment

Additional comments

The manuscript is concerned with the analyse the impact of the land drainage measures implemented under the framework of the above program on assemblages of ground beetles. This is very interesting study which try to assess the changes in Carabid community under land drainage works. The study was conducted in 2009, 2012 and 2013 in area formerly used for farming and then afforested and turfed. The authors choose two study sites differed by moisture conditions and distance form the retention water reservoirs. Ground beetles were collected in from three transects of pitfall traps set in study site I and three in study site II. I think this study is interesting and it is important to know the shift in ground beetle species diversity in such conditions especially in case of protection of rare and valuable species. However there are some obscurity. My recommendation for the current submission is to Minor revision. A list of comments with line numbers is given below.

Annotated reviews are not available for download in order to protect the identity of reviewers who chose to remain anonymous.

Reviewer 2 ·

Basic reporting

The work is, unfortunately, written in very inadequate English, with several important terms erroneously used (e.g. for feeding type or trophic group, the word "trophy" is used) that leaves the reader grasping for meaning.
The conceptual background is murky, and there is considerable confusion about water retention (which is not drainage), as well as what exactly carabids will indicate in this context. The aims of the work are not clear.
As there are no specific hypotheses, results are not well structured, and the impression is that the authors themselves are unsure what precisely their findings mean? This leads to very simple and qualitative statements, like the presence of ca. 30% of the regional fauna, the appearance of moisture-preferring species, stb.
Diversity analysis is restricted to a very simple and indequate method.
Overall, these do not convince the reader that there is much new in this work.

Experimental design

There is no real repetition here - only one setup is studied, and the trapping effort is decidedly modest (15 traps). Strange terminology makes me a bit unsure about important details (e.g. what is "sphere"?)
The research question is not well defined, it seems like merely a "let's see what happened" type.
The diversity evaluation is very superficial, and there is no test about the completeness of the study - for example, by species accumulation curves, or an estimation of the number of species present.
The claimed information gap is only the scarce studies of peatlands of the study region. No justification is given why does this matter?

Validity of the findings

Not a big study with insufficient replication. This undermines the validity.
No well-formulated research questions, thus "results" leave the reader uninterested, and makes difficult to draw meaningful conclusions. In fact, those are very non-specific and qualitative. Imprecise and erroneous terminology adds to the confusion (re. drainage vs. water retention).

Additional comments

This work aimed to examine the changes in carabid assemblages triggered by the establishment of a water retention scheme. Unfortunately, the language is so bad that in many places, the reader has to guess the intended meaning. You will have to ask for help from a colleague with a better command of English.
You use the term community vs. assemblage as if they were equivalent. I suggest that your study concerns a taxonomic group and not an ecological one, so I'd use assemblage throughout. The correct term then would be "ground beetle assemblage/s" or "carabid assemblage/s".
Your trapping intensity is not high - at minimum, you have to prove its completeness by the use of a species-accumulation curve. Diversity evaluation is very primitive - use the scheme suggested by Henderson & Southwood's Ecological methods (2016, Wiley).
You also have to think about indication - what do you want to indicate and which parameters of the carabid assemblage would indicate that? This is not mentioned nor elaborated.

Reviewer 3 ·

Basic reporting

The manuscript is written in good English. The introduction provides with a sufficient overview of the scientific background of the paper with well-chosen references. The structure of the manuscript is according to the instructions given by PeerJ.

All table and figures are relevant for the paper and the figures are of good graphical quality. However, the captions of figs. 2 and 4 have to be improved. In the caption to fig. 2 lacks the information what is shown by the error bars. Did you calculate standard deviation or standard error? This is an important information. The letters “a”, “b, and “c’ should be in capitals as in the figure itself. Please add also the unit for MIB, which is mg. In the caption to fig. 4 it would be helpful for the reader to add at least a reference to the methods with respect to the abbreviations of the ecological groups.

According to the PeerJ policy the raw data are provided with the manuscript.

Experimental design

The manuscript is based on original primary research, which falls into the general scope of the journal. The basic purpose of the study is well defined (lines 66-69), but the authors might formulate some expectations (research hypotheses). Which kind of changes in carabid assemblages are expected? Which differences regarding the changes are expected between the different studied spheres?

The research was carried out accurately according to standards in the research on ground beetles (Carabidae).

The methods description should be improved. Information about the basic samples used in statistical analyses is missing. The authors mention that there were 15 traps in each object (5 per transect = sphere), which were emptied every 14 days. However, it is not clear if the material from the14-day-intervals were later pooled for each trap and if each trap was used as sample. Unfortunately the supplemental files “…raw_data.csv” and “…Statistical_data.csv” also give no precise information concerning this matter. The file, “Statistica_data.csv” contains several lines with data for each trap in each year (for example nine for the year 2009). Thus, I assume that these are nine 14-day-intervals. However, this should be indicated in the file. The file “…raw_data.csv”, however, shows only 5 columns for zone A in 2009. I assume that this are the data for the individual traps pooled. Instead of using “A”, “B”, etc. in this table I propose to use the respective trap numbers. From the file “…Statistical_data.csv” I would conclude that each 14-day-interval was used as basic sample in the statistics, but from fig. 4 I would conclude that the pooled data were used. Therefore, it has to be precisely described in the methods which basic samples were used for which statistical analysis.

I also suggest to provide more details regarding the PCA analysis (lines 159-162). Why was a linear method (PCA) selected instead of an unimodal method (CA)? Which settings (for example regarding data transformation) were used in the analyses?

Validity of the findings

The manuscript deals with an interesting and important topic. The authors accentuate with indication of reference that studies on entomofauna inhabiting areas submitted to land drainage programs are rare, hence underlining the importance of the research for a wider audience. The underlaying data seem to be robust and statistically sound. However, as mentioned above, I recommend some improvements in the supplemental files.

Results are described accurately, but I like to encourage the authors to rethink the description of the results of the PCA (lines 219-229). Even if I agree with the general trends formulated by the authors, the results are complex and not all assemblages of the respective years follow the described trends entirely. Therefore, I recommend to emphasize that the results have more tendency character. I propose also to emphasize the MIB indicator, since there is a clear tendency that assemblages from 2009 are characterized by lower MIB values, what is in accordance with the dominating of large zoophages after the retention project, as stated by the authors.

The results are discussed in a though-out manner taking into account an adequate number of references.

The conclusions formulated by the author are well stated. One or two sentences regarding possibilities of practical application of the results might be added.

Additional comments

Line 43: Probably it should be “mentioned” instead of “maintained”. Please check.

Line 55: The reference “Kosewska et al., 2015” is not included in the reference list. Instead, there is a reference “Kosewska & Nietupski, 2015”. Please specify.

Line 146: Has to be “Pacuła” instead of “Pacuk”.

Line 524: Please add information when the website was accessed.

Line 525: The reference “Zabrocka-Kostrubiec, 2008” should be shifted to the end of the reference list.

---

## Round 0.2 · Minor Revisions

Dear Dr. Ludwiczak and colleagues:

Thanks for revising your manuscript. The reviewers are very satisfied with your revision (as am I). Great! However, there are a few minor edits to make. Please address these ASAP so we may move towards acceptance of your work.

Best,

-joe

Reviewer 1 ·

Basic reporting

All my comments have been adressed.
One note: I found a small literal erros in the reference Igondová & Majzlan, 2015 (line 241 is "Igondová" and line 432 is "Igondowá" please improve it.

Experimental design

All my comments have been adressed.

Validity of the findings

All my comments have been adressed.

Additional comments

Thank the Authors for revising the manuscript. All my comments have been taken into account, so my recommendation for the current submission is to accept to publication.

Reviewer 3 ·

Basic reporting

Formal flaws of the earlier version of the manuscript are corrected in the revised version. Due to the adding of research hypotheses the goals of the study are more clear now. The replacement of the PCA by an RDA clearly improves the manuscript.

Experimental design

The research design still could be better explained in the methods. The authors state in the response to my review that “each trap was treated as a replication. This information has been added to the manuscript text”. However, I cannot find this information in the revised manuscript! If I understand well, each trap as a replication was used for GLM analysis (Table 2), mean values of species richness, abundance and MIB (Figure 2), NMDS (Figure 4) and RDA (Figure 5). The authors should clearly state it. On the other hand, if I understand well, for calculation of species numbers, individual numbers and Shannon Diversity (Table 3) and for the species accumulation curves (Figure 3) the data of the five traps of each transect were pooled. This should be also stated in the text.

Validity of the findings

Instead of two somewhat differing supporting files, the authors provide now with one file, which shows in a sufficient manner the full set of raw data.

---

## Round 0.3 · accepted · Accept

Dear Dr. Ludwiczak and colleagues:

Thanks for revising your manuscript based on the concerns raised by the reviewers. I now believe that your manuscript is suitable for publication. Congratulations! I look forward to seeing this work in print, and I anticipate it being an important resource for groups studying environmental impacts on ground beetles. Thanks again for choosing PeerJ to publish such important work.

Best,

-joe